# Analysis of Polyvinylidene Fluoride Membranes Fabricated for Membrane Distillation

**DOI:** 10.3390/membranes11060437

**Published:** 2021-06-10

**Authors:** Minchul Ahn, Hyeongrak Cho, Yongjun Choi, Seockheon Lee, Sangho Lee

**Affiliations:** 1School of Civil and Environmental Engineering, Kookmin University, Seoul 02707, Korea; 92vet@naver.com (M.A.); rhino@kookmin.ac.kr (H.C.); choiyj1041@gmail.com (Y.C.); 2Water and Resource Recycle Center, Korea Institute of Science and Technology, Seoul 02792, Korea; seocklee@kist.re.kr

**Keywords:** membrane distillation, membrane fabrication, analysis, flux, wetting, liquid entry pressure, contact angle

## Abstract

The optimization of the properties for MD membranes is challenging due to the trade-off between water productivity and wetting tendency. Herein, this study presents a novel methodology to examine the properties of MD membranes. Seven polyvinylidene fluoride (PVDF) membranes were synthesized under different conditions by the phase inversion method and characterized to measure flux, rejection, contact angle (CA), liquid entry pressure (LEP), and pore sizes. Then, water vapor permeability (*B_w_*), salt leakage ratio (*L_w_*), and fiber radius (*R_f_*) were calculated for the in-depth analysis. Results showed that the water vapor permeability and salt leakage ratio of the prepared membranes ranged from 7.76 × 10^−8^ s/m to 20.19 × 10^−8^ s/m and from 0.0020 to 0.0151, respectively. The *R_f_* calculated using the Purcell model was in the range from 0.598 μm to 1.690 μm. Since the *R_f_* was relatively small, the prepared membranes can have high LEP (more than 1.13 bar) even at low CA (less than 90.8°). The trade-off relations between the water vapor permeability and the other properties could be confirmed from the results of the prepared membranes. Based on these results, the properties of an efficient MD membrane were suggested as a guideline for the membrane development.

## 1. Introduction

The growth of world population and industry has posed challenges associated with the imbalance between water demands and availability. The situation has been worsened by the impact of climate change, which results in unexpected fluctuations in rainfall patterns and the availability of water resources [1,2]. One of the technologies that alleviate this problem is desalination of seawater, brackish water, or wastewater [3,4]. Since desalination technique does not rely on freshwater resources, it can provide ongoing water supply by utilizing alternative water sources [5]. This is the key driver that makes the widespread adoption of desalination technology over the past decades [6,7,8].

However, the implementation of desalination requires a viable option for the management of brines, which is the remaining stream after the production of fresh water from seawater or impaired water sources [9,10]. Conventional desalination processes including multistage flash (MSF), multi-effect distillation (MED), and reverse osmosis (RO) result in a substantial amount of brines ranging from 40% to 60% of the feed water [11]. Since the brine exhibits a higher salinity than the feed water, its direct discharge may lead to harmful effects on marine environment and aquatic ecosystems [12]. Accordingly, it is crucial to explore novel techniques to reduce the adverse impact by reducing the volume or salinity of the brine [12].

A promising option for brine management is to develop and apply membrane distillation (MD) process that allows the reduction of the brine volume and the additional production of fresh water [13,14,15]. MD is a special evaporation technique and requires hydrophobic microporous membranes to separate feed water and vapors [16,17,18]. Since MD is thermally driven, its operation is not limited by the osmotic pressure, allowing the treatment of high salinity brine from RO or MSF/MED processes [19,20]. Moreover, MD may use moderate-temperature thermal sources (50–70 °C) that are cheaper than high-temperature heat sources used by MSF/MED [21]. MD can be considered as a part of zero liquid discharge (ZLD) systems [22]. Many MD systems for brine management have been reported in the literature due to their potential [17,23,24]. Nevertheless, most works were done in laboratory scales [8,25].

One of the biggest hurdles MD technology faces is the availability of efficient MD membranes [26,27]. There are several commercial hydrophobic membranes for MD experiments but they are not specifically optimized for MD [23]. This is because the requirements for MD membranes are quite unique [26]. Pore wetting is a critical issue in MD processes, which affects the quality of product water. To minimize it, it is necessary to fabricate hydrophobic MD membranes or increase the liquid entry pressure (LEP) [24,28]. On the other hand, fouling and scale formation are also serious problems that reduce the flux and lifespan of MD membranes. The pretreatment of feed water as well as physical/chemical cleaning may be applied to mitigate the adverse impact of fouling and scaling [29]. Nevertheless, it is also important to improve the fouling resistance of MD membranes by surface modifications [29]. To prevent adsorptive fouling caused by hydrophobic organic matters, MD membranes should be rather hydrophilic. The pore size of the MD membranes should be as small as possible to decrease the possibility of pore wetting. On the contrary, the flux and water productivity of the MD membrane increase with an increase in the pore size [15,16]. Although numerous works have been done on the synthesis of MD membranes [28,30,31,32,33,34,35,36], it is still challenging to optimize its conditions due to the complex trade-offs.

Accordingly, it is necessary to implement systematic approaches to meet different requirements of MD membranes. However, only limited information is available for in-depth analysis of MD membrane characteristics. In addition to flux and rejection, more fundamental performance indexes should be used to provide insight into the multi-objective optimization of synthesis conditions for MD membranes. Herein, a novel methodology to examine the properties of MD membranes was developed and applied to several polyvinylidene fluoride (PVDF) membranes synthesized under different conditions by the phase inversion method. In previous works, protic or aprotic solvents such as methanol, propanol, butanol, octanol, and N-methyl-prrolidone were added to the non-solvent, leading to an increase in the hydrophobicity of the membrane surface [37,38,39]. However, these solvents are toxic, and pose threats to the environment and human health. Therefore, water and ethanol were used in this study because they are more environmentally friendly. The prepared membranes were characterized to determine flux, rejection, contact angle (CA), liquid entry pressure (LEP), and pore sizes. Then, a set of new performance measures such as water vapor permeability (*B_w_*), salt leakage ratio (*L_w_*), and fiber radius (*R_f_*) were proposed. The correlations between different membrane properties were also investigated to understand the trade-offs in MD membranes. Based on this analysis, the properties of efficient MD membranes were explored.

## 2. Materials and Methods

### 2.1. Materials

PVDF was purchased from Sigma-Aldrich Co. (St. Louis, MO, USA) and its molecule weight was 530,000 g/mol. N,N-Dimethylformamide (DMF, 99.9%), lithium chloride (LiCl, 98.2%), and ethyl alcohol (EtOH, 99.9%) were supplied by Samchun Inc. (Gyeonggi-do, Korea). Deionized (DI) water was obtained using a water deionizer (HUMAN POWER, Human Co., Seoul, Korea). All chemicals were used without further purification in this work.

### 2.2. Fabrication of PVDF Membranes

To begin, PVDF solutions of different concentrations were prepared by dissolving PVDF powder and LiCl in DMF solvent and stirring at 300 rpm for 3 h at 80 °C [40], then the solution was stood for 24 h to remove air bubbles. The prepared PVDF solution was transferred on a flat glass plate. A casting machine (motorized film applicator, Elcometer Inc., Manchester, UK) and casting knives (casting knife film applicator, Elcometer Inc., Manchester, UK) were used to control the thickness of the covered film at 300 μm. The covered film solution was immediately soaked in a coagulation bath containing DI water and/or EtOH for 1 h. As the final step, the membrane was placed in an oven at 60 °C for 24 h to obtain a dried, flat-sheet of membrane. By varying the compositions of the PVDF solution and the non-solvents, 7 different PVDF membranes were fabricated. The conditions of the membrane fabrication are summarized in Table 1.

### 2.3. Contact Angle (CA) Measurement

The technique of sessile drop contact angle measurement was applied to the fabricated membranes as previously reported [11,40]. An instrument to measure the contact angle (Smart Drop) was supplied by Femtobiomed (Gyeonggi-do, Korea). The following procedures were used: (1) Membrane samples were placed on a plate; (2) water droplets (5 μL) were placed onto the membrane surface; (3) after the stabilization, the camera in the device captured images of the droplet from five different positions; and (4) the image analysis software connected to the instrument automatically determined the CA from the images. The measurements were repeated at least seven times per membrane sample and the average and standard deviation were recorded.

### 2.4. Measurement of Liquid Entry Pressure

An in-house apparatus was used to measure the liquid entry pressure (LEP) of the fabricated membranes [11,40]. As shown in Figure 1, the apparatus consists of an LEP chamber, a pressure gauge, a high-pressure N_2_ gas cylinder, and a pipe. To begin, 50 mL DI water was poured into the chamber and a dried membrane sample was mounted. The chamber was then pressurized from its bottom using the N_2_ gas cylinder connected with the pipe. The pressure was controlled using the gauge to determine the minimum pressure that resulted in the first water drop on the membrane surface. The LEP measurements were repeated three times and the average and the standard deviation were recorded.

### 2.5. Analysis of Membrane Morphology

A FE-SEM (FE-SEM 7800F Prime, JEOL Ltd., Tokyo, Japan) instrument was used for observing the microstructures of the prepared membranes as previously reported [41]. Before the analysis, the membrane samples were completely dried at 60 °C for 2 h in a drying oven. Then they were coated with platinum for 30 s by sputtering.

Atomic Force Microscope (AFM) was applied for an in-depth analysis of membrane surface. An AFM instrument (AFM, Atomic Force microscope, XE-100, PSIA Inc., Gyeonggi-do, Korea) was used to obtain the images. Based on the measurement results, the arithmetical mean deviation (*R_a_*), root mean square deviation (*R_q_*), and the vertical distance between the highest peak and lowest valley (*R_max_*) were calculated using the same scan size (5 μm × 5 μm).

### 2.6. Analysis of Pore Size Distribution and Thickness Measurement

A capillary flow porometer (CFP-1500-AFL, porous materials Inc., Ithaca, New york, USA) was employed to measure the pore size distribution of the membranes [40]. Prior to the analysis, the membrane samples were soaked into the Galwick solution (porous materials Inc., Ithaca, New york, USA, surface tension = 15.9 dynes/cm), then N_2_ gas was applied to the wetted membrane samples to obtain raw data required for the calculation of the pore size distribution of the membranes. The mean pore diameter (*d_mean_*), maximum pore diameter (*d_max_*), and the minimum pore diameter (*d_min_*) were also estimated. The thickness of the membranes was measured using digital vernier calipers (Mitutoyo Inc., Kawasaki, Japan). At least three different locations on each sample were selected and analyzed to confirm the uniformity of the membranes, and the average and standard deviation were provided.

### 2.7. Measurement of Porosity

The gravimetric method was adopted to estimate the porosity of the membranes [42]. First, the membrane samples of equal size (2 × 2 cm^2^) were immersed in ethanol. Second, the weights of the samples before and after saturation with ethanol were compared. Using the measured data, the membrane porosity (*ε*) was calculated:(1)$$\varepsilon =\frac{\left(m_{1}-m_{2}\right)/D_{e}}{\left[\left(m_{1}-m_{2}\right)/D_{e}+m_{2}/D_{p}\right]}$$
where *m*_1_ is the mass of the saturated membrane (g), *m*_2_ is the mass of the dry membrane (g), *D_e_* is the specific gravity of the ethanol (g/cm^3^), and *D_p_* is the specific gravity of the PVDF material (g/cm^3^).

### 2.8. Measurement of Flux and Rejection

The flux and salt rejection of the membranes were determined by carrying out a set of direct contact membrane distillation (DCMD) experiments. The schematic diagram of the DCMD set-up is illustrated in Figure 2. Details on this technique were previously reported [40,41] and only a slight modification was done in this work. The effective membrane area was 12 cm^2^. The initial volumes of the feed and the permeate were 2.0 L and 1.0 L, respectively. The feed and permeate temperatures were fixed at 60 ± 1.5 °C and 20 ± 1.5 °C, respectively, which were controlled using a heater and a chiller. The flow rates of the feed and permeate were 0.7 L/min and 0.4 L/min, respectively. Moreover, the feed and permeate pressures were measured to be 0.08 bar and 0.02 bar, respectively.

The feed solution was 35 g/L NaCl solution and the permeate solution was the DI water. The electrical conductivity of the feed and permeate was measured using a conductivity meter (WTW multi 3420, WTW, Munich, Germany). The conductivity was converted to the concentration using a standard curve [11,41]. The weight of the permeate tank was periodically measured using an electronic balance (Explorer Pro, Ohaus, Newark, NJ, USA) and the flux was calculated based on the following equation [13,43]:(2)$$J_{v}=\;\frac{\Delta m_{g}}{A_{m}\Delta t}$$
where Δ*m_g_* is the increased mass of the permeate, *A_m_* is the membrane areas, and Δ*t* is the time interval. The apparent salt rejection (*R_app_*) was calculated by [44]:(3)$$R_{app}=1-\frac{c_{p}}{c_{f,0}}$$
where *c_f,_*_0_ is the initial feed concentration, and the *c_p_* is the permeate concentration. It should be noted that the intrinsic rejection (*R_int_*) is different from the apparent rejection in the DCMD operation. Accordingly, *R_app_* may be misleading because it generally overestimates the rejection capability of the membrane. The technique to calculate *R_int_* from *R_app_* will be discussed later.

### 2.9. Calculation of Additional Membrane Properties

The flux, rejection, contact angle (CA), liquid entry pressure (LEP), and pore sizes are primary properties of MD membranes. However, they are not sufficient to understand the complicated trade-offs of MD membrane performances. The flux and rejection are apparent properties and affected by the operating conditions. The CA and LEP cannot be directly correlated if the membrane morphology is complex. Accordingly, a set of equations were derived and used to evaluate additional (secondary) properties, including water vapor permeability (*B_w_*), salt leakage ratio (*L_w_*), and fiber radius (*R_f_*).

#### 2.9.1. Calculation of Water Vapor Permeability (*B_w_*)

The water vapor transport in the DCMD process is driven by the difference in the vapor pressure between both sides of the membrane. The dependence of the water flux (*J_v_*) on the vapor pressure difference (Δ*p_w_*) is described by [18,19]:(4)$$J_{v}=B_{w}\Delta p_{w}=B_{w}\left(p_{w,f}^{0}a_{w,f}-p_{w,p}^{0}a_{w,p}\right)\approx B_{w}p_{w,f}^{0}a_{w,f}=B_{w}p_{w,f}^{0}\gamma_{w,f}x_{w,f}$$
where *B_w_* is the water vapor permeability; *p_w_* is the vapor pressure; *a_w_* is the activity; *γ* is the activity coefficient; *x_w_* is the mole fraction of water. The subscripts *f* and *p* refer to feed and permeate, respectively, and the superscript 0 refers to pure water. *B_w_* is an intrinsic coefficient depending on the membrane properties such as pore diameter, porosity, and the length of the pores, but also affected by the applied temperature.

The vapor pressure of pure water is estimated by the Antoine equation [45]:(5)$$p_{w,f}^{0}=e^{\left(23.1964-3816.44/\left(T_{f}-46.13\right)\right)}$$

When NaCl is used to prepare the feed solution, the activity coefficient of the water in the feed (*γ_w,f_*) is given as a function of the mole fraction of NaCl (*x_NaCl_*) [16,45]:(6)$$\gamma_{w,f}=1-0.5x_{NaCl}-10x_{NaCl}^{2}$$

The mole fraction of the water in the feed is [16,45]:(7)$$x_{w,f}=1-x_{NaCl}$$

Combining the Equations (4)–(7), the *B_w_* is calculated from experimentally-determined flux by use of:(8)$$B_{w}=\frac{J_{v}}{\left(e^{\left(23.1964-3816.44/\left(T_{f}-46.13\right)\right)}\right)\left(1-0.5x_{NaCl}-10x_{NaCl}^{2}\right)\left(1-x_{NaCl}\right)}$$

*B_w_* corresponds to water permeability of a membrane in pressure-driven membrane processes such as MF, UF, and RO. Since it does not change by the feed concentration (i.e., *x_NaCl_*), it may be used as an intrinsic property of a membrane.

#### 2.9.2. Estimation of Salt Leakage Ratio (*L_w_*)

The salt transport in the DCMD process (*J_s_*) is attributed to two mechanisms: (1) the transfer of salts with the water vapor (distillation) and (2) the leakage of feed water through wetted pores. In theory, the distilled water does not contain any salts and the contribution of the first mechanisms is negligible. Accordingly, the following equation is obtained to calculate *J_s_*:(9)$$J_{s}=\left(\frac{J_{v}}{\rho }\right)\left(1-L_{w}\right)c_{p,distill}+\left(\frac{J_{v}}{\rho }\right)L_{w}c_{p,leak}\approx \left(\frac{J_{v}}{\rho }\right)L_{w}c_{p,leak}=\left(\frac{J_{v}}{\rho }\right)L_{w}c_{f}=\left(\frac{J_{v}}{\rho }\right)c_{p,net}$$
where *L_w_* is the leakage ratio, which is defined as the ratio of water leakage to total permeate; *ρ* is the density of water, *c_p,distill_* is the salt concentration in the distilled water; *c_p,leak_* is the salt concentration in the water leakage; *c_p,net_* is the salt concentration of the water transferred through the membrane; *c_f_* is the salt concentration in the feed. It should be noted that the *c_p,net_* is different from the apparent permeate concentration (*c_p_*) since the water transferred through the membrane is mixed with the recirculated water in the permeate side. Accordingly, *c_p,net_* = *L_w_ c_p,leak_* = *L_w_ c_f_* ≠ *c_p_*. This leads to a difference between the apparent rejection (*R_app_*) and the intrinsic rejection (*R_int_*), which is given by:(10)$$R_{\mathrm{int}}=1-\frac{c_{p,net}}{c_{f}}=1-\frac{L_{w}c_{f}}{c_{f}}=1-L_{w}$$

In DCMD processes, the mass balances of the salt and the water can be established respectively as follows:(11)$$\frac{d\left(c_{p}V_{p}\right)}{dt}=\left(\frac{J_{v}}{\rho }\right)L_{w}c_{f}A_{m}$$
(12)$$\frac{dV_{p}}{dt}=\left(\frac{J_{v}}{\rho }\right)A_{m}$$

Accordingly, the following equation is used to calculate *L_w_* from the results of DCMD experiments:(13)$$L_{w}=\left(\frac{\rho }{J_{v}}\right)\frac{\frac{d\left(c_{p}V_{p}\right)}{dt}}{c_{f}A_{m}}=\frac{\frac{d\left(c_{p}V_{p}\right)}{dt}}{c_{f}\frac{dV_{p}}{dt}}\approx \frac{\frac{\Delta \left(c_{p}V_{p}\right)}{\Delta t}}{c_{f}\frac{\Delta V_{p}}{\Delta t}}$$

#### 2.9.3. Determination of Fiber Radius (*R_f_*)

The Young-Laplace model is a well-known equation to correlate LEP with contact angle. With the assumption of cylindrical pores, LEP is described by [46]:(14)$$\Delta P=\left(\frac{-2\gamma }{r}\right)\cos\theta$$
where Δ*P* is liquid entry pressure, *γ* is the surface tension of the feed solution, *r* is pore radius, and *θ* is the intrinsic contact angle between the liquid and the membrane material [47]. However, the Young-Laplace model does not reflect the effect of membrane surface morphologies and cannot explain positive LEP values for membranes with *θ* less than 90°, which have been reported in the literature. In this case, the Purcell model (Figure 3) should be applied instead of the Young-Laplace model since it allows the prediction of positive LEP for membranes with relatively small contact angle. The equation for LEP in the Purcell model is given by [47,48]:(15)$$\Delta P=\left(\frac{-2\gamma }{r}\right)\frac{\cos(\theta +\alpha )}{1+\frac{R_{f}}{r}(1-\cos(\alpha ))}$$
(16)$$\sin(\theta +\alpha )=\frac{\sin\theta }{1+\frac{r}{R_{f}}}$$
where *R_f_* is the fiber radius and α is the angle below horizontal in the fiber. The *R_f_* is calculated by simultaneously solving Equations (15) and (16). It is worth noting that *R_f_* is an important intrinsic property of the MD membranes. Although the surface properties are similar, the LEP of the membranes changes with different *R_f_* values. In other words, the LEP, which is related to wetting resistance, is affected not only by the hydrophobicity of the material but also by morphological parameters such as *R_f_*.

## 3. Results and Discussion

### 3.1. Characterization of Fabricated Membranes

As indicated in Table 1, seven membranes were fabricated under different conditions. The PVDF concentration, LiCl concentration, and the composition of the non-solvent were varied. The surface and cross-sectional images of the prepared membranes were presented in Figure 4 and Figure 5. As a common observation, all the membranes were asymmetric and showed both finger-like and sponge-like regions. Smaller pores seem to be formed in the membranes prepared without LiCl (S1, Figure 4a and Figure 5a) than the other membranes. The addition of LiCl to the casting solution increased the rate of precipitation during the immersion step of phase inversion process, leading to the formation of a coarser structure of the membrane [39]. When the PVDF concentration increased from 14 wt.% to 18 wt.% in the presence of LiCl (S2, S3, S4, Figure 4b–d and Figure 5b–d), the sizes of the finger-like structures were reduced as a result to the retardation of the phase separation rate [34,49]. On the other hand, the addition of EtOH to the non-solvent (S5, S6, S7, Figure 4e–g and Figure 5e–g) decreased the sizes of the finger-like structures and enlarged the sponge-like regions. This is attributed to a reduction in the phase separation rate as an increase in the EtOH concentration [49]. The solubility parameter for PVDF and EtOH is smaller than that for the PVDF and water due to the effect of the hydrogen bonding [50], indicating that PVDF is more miscible with EtOH than water. Accordingly, a slower phase separation is expected in the presence of EtOH in the non-solvent.

Table 2 summarizes the characteristics of the fabricated membranes. The contact angles of the fabricated membranes ranged between 75.1° and 90.3°. The addition of LiCl to the casting solution does not seem to significantly affect the contact angle (S1, S2). The contact angle did not increase with an increase in the PVDF concentration from 14.0 wt.% to 18.0 wt.% (S2, S3, S4). One the other hand, the contact angle increased by increasing the EtOH concentration in the non-solvent from 0 wt.% to 30 wt.% (S3, S5, S6, S7). Since the addition of EtOH instead of water retarded the rate of phase separation, the size of the polymer crystals increased as well as the surface roughness, thereby increasing the contact angle [26,39]. Although the contact angles were relatively small, the LEP values were higher than generally expected. According to the Young-Laplace equation, the membranes with the contact angle less than 90° should have negative LEP values. However, the LEP values for the fabricated membranes were measured in the range between 1.13 bar to 3.19 bar. Without LiCl, the LEP value was the highest (3.19 bar, S1), which is attributed to its dense structures. In the presence of LiCl, the LEP increased with an increase in the PVDF concentration (S2, S3, S4). It should be noted that the LEP values were different with similar values of the contact angle in these cases. As the EtOH concentration in the non-solvent increased, the LEP increased and then decreased (S3, S5, S6, S7). Again, the LEP and contact angle did not exhibit a strong correlation.

As presented in Table 2, the membrane thickness varied in the range from 68.0 μm to 98.8 μm. The thinnest membrane was obtained in the absence of LiCl (S1) while the thickest membrane was prepared with the highest PVDF concentration (S4). In the other cases, the thickness values were similar. The porosity of the membranes ranged from 81.9% to 88.3%. Although the morphologies of the membranes were different as shown in Figure 4 and Figure 5, the porosities were not significantly different. This suggests that the size and shape of the void space in the membranes are different even with the similar porosity.

The mean (average), maximum, and minimum diameters of the pores in the membranes are presented in Table 2. The mean pore diameter of the membrane prepared without LiCl (S1) was 0.09 μm, which was the smallest. The mean pore size increased after adding LiCl. Moreover, the maximum and minimum pore diameters showed an increment by up to 2.38 times (S7) and 1.71 times (S2), respectively. As presented in Figure 6a, the pore size distribution shifted to the right by adding LiCl. However, no evident dependence of the pore size distribution on the PVDF concentration was observed. With an increase in the EtOH concentration from 10 wt.% to 30 wt.%, the pore size distribution moved to the right, as illustrated in Figure 6b.

### 3.2. DCMD Performance

Using the prepared membranes, a set of DCMD experiments were carried out in the experimental set-up (shown in Figure 2). The results are presented in Figure 7. The black symbols indicate the permeate flux and the red ones point out the permeate concentration of NaCl. Each DCMD experiment was carried out for 24 h. As shown in Figure 7a, the membrane prepared without LiCl (S1) resulted in a flux less than 6 kg/m^2^-h and a permeate concentration less than 6 mg/L. The fabrication of the membranes in the presence of LiCl (S2, S3, and S4) substantially increased both the flux and permeate concentration. When the PVDF concentration was 14 wt.%, the increasing rate of the permeate concentration was the highest (Figure 7b). This is attributed to the fact that S2 had a wide pore size distribution and low LEP as shown in Table 2. A lower LEP may be related to a higher probability of salt passage, leading to higher permeate concentration. It gradually decreased with an increase in the PVDF concentration (Figure 7c,d). The increased concentration of EtOH in the non-solvent (S5, S6, S7) also affected the flux and permeate concentration as presented in Figure 7e–g.

The average flux and the apparent rejection in the DCMD experiments are summarized in Table 3. It is evident that there is a trade-off between flux and rejection. The membrane with the highest rejection showed the lowest flux (S1) while the membrane with the highest flux exhibited the lowest rejection (S2). Nevertheless, there was also an exception that allows relatively high flux and rejection simultaneously (S5, S7). Figure 8 gives an overview of the effect of fabrication conditions on the flux and rejection of the MD membranes. Since the focus of this work is not on the optimization of the membrane fabrication, the performance of these membranes such as flux and rejection may not be optimized. Nevertheless, it should be noted that the properties of the membranes are sensitive to their fabrication conditions, leading to different performances. Accordingly, an in-depth analysis to understand the relations among the membrane properties is required.

### 3.3. In-Depth Analysis of Membrane Properties

To further investigate the characteristics of the MD membranes, the “secondary” properties were calculated using the Equations (8), (13), (15), and (16). The results are presented in Table 4.

The average water vapor permeability (*B_w_*) ranged from 7.76 s/m to 20.19 s/m. As expected, a membrane with a high flux showed a high *B_w_*. The salt leakage ratio (*L_w_*) was observed in the range from 0.0020 to 0.0151. A membrane having a high rejection resulted in a low *L_w_*. Since *B_w_* and *L_w_* are less dependent on the experimental conditions (i.e., feed concentration, MD operation time) than flux and rejection, they may be used as intrinsic properties of a membrane. Nevertheless, care should be taken because they may be also affected by the conditions such as feed temperatures.

Table 4 also presents fiber radius (*R_f_*), which is related with the morphology of the membranes. Membranes prepared by the phase inversion technique often have pores consisting of the spaces between individual membrane fibers. *R_f_* is the radius of the membrane fibers. In the Equations (15) and (16), LEP decreases as an increase in *R_f_* if all the other conditions are the same. This suggests that membranes with the same pore size and hydrophobicity may have different LEP due to different *R_f_* values. The *R_f_* value of the membranes was calculated in the range from 0.598 μm to 1.690 μm.

### 3.4. Correlations among Different Properties

As the next step, the correlations between different membrane properties were investigated to explore the way to fabrication of efficient MD membranes. Figure 9a reveals the relationship between *B_w_* and *L_w_*. As expected, an increase in *B_w_* resulted in an increase in *L_w_*, indicating a trade-off between the two properties. This is attributed to the fact that the membranes with higher *B_w_* values have larger pore sizes. As presented in Figure 9b, the mean pore radius increased with an increase in *B_w_*. Since *L_w_* is related to the partial pore wetting, the membranes with larger pore sizes may have higher *L_w_* values. The dependence of *R_f_* on *B_w_* is presented in Figure 9c. It is evident the *R_f_* increases with an increase in *B_w_*, implying that it is difficult to obtain membranes with high *B_w_* and small *R_f_*. This is probably because the membranes with large *R_f_* may have larger pores due to their coarse structures.

To provide insight into the factors affecting LEP, the effect of contact angle and pore size on LEP was further analyzed. As presented in Figure 10a, there was no correlation between the contact angle and LEP. Although LEP is affected by chemical properties represented by the contact angle, it is also a function of physical properties such as surface morphology. At least in our case, the latter seems to be more important than the former. On the other hand, the LEP is inversely proportional to the pore diameters as shown in Figure 10b. Not only the maximum pore diameter (*d_max_*) but also the mean pore diameter (*d_mean_*) and minimum pore diameter (*d_min_*) showed reasonable correlations. A reduction in LEP by an increased pore size can be explained by the Equations (15) and (16).

As mentioned before, it is desired to obtain membranes with high *B_w_* (or high flux) and high LEP. Based on the Equations (15) and (16), LEP increases with an increase in pore size and a decrease in *R_f_*. Accordingly, the control of *R_f_* may be a novel approach to increase LEP without sacrificing flux. Unfortunately, it seems that there is also a trade-off between the pore size and *R_f_*. As presented in Figure 11, *R_f_* increased as the pore size increased. This is probably because small membrane fibers are required to form small pores. Nevertheless, a further investigation is recommended to explore the method to increase the pore size and decrease *R_f_* simultaneously.

Although *R_f_* cannot be directly measured, it may be related to measurable properties of the membranes. To explore the possibility, AFM analysis was performed to determine the surface roughness parameters, which are used to correlate with *R_f_*. The 3-D surface images of membranes are presented in Figure 12. Using the images, the arithmetical mean deviation (*R_a_*), root mean square deviation (*R_q_*), and vertical distance between highest peak and lowest valley (*R_max_*) were calculated and summarized in Table 5. As a general observation, the roughness parameters increased with an increase in the EtOH concentration. Although S3 (0 wt.%) and S5 (10 wt.%) showed similar roughness parameters, S6 (20 wt.%) and S7 (30 wt.%) clearly exhibited higher values of the roughness parameters. This may be attributed to the occurrence of solid-liquid demixing (crystallization) in the presence of EtOH.

Figure 13 presents the dependence of the roughness parameters on *R_f_*. There are increasing tendencies of the roughness parameters on *R_f_*, suggesting that *R_f_* may be related to the measurable properties. It is also plausible that an increase in *R_f_* increases the surface roughness due to an increase in the distance between highest peak and lowest valley on the membrane surface.

Table 6 lists membrane properties between the current work and the previous investigation [40]. Although the contact angle of the commercial PVDF membrane was the highest (126°), the LEP was the lowest among these membranes. This is attributed to the fact that its pore size is the largest. The M1 and M2 membranes showed low contact angles but their LEP values were high due to their relatively small pore size as well as small *R_f_* values. On the other hand, the S6 and S7 membranes showed moderate contact angles and high LEP values. The apparent rejection of all membranes was similar, but the salt leakage ratio was different, ranging from 0.0015 (M1) to 0.0045 (S7). The *R_f_* ranged from 0.350 (M2) to 1.503 (S7) membranes.

## 4. Conclusions

This study presents an approach to evaluate the properties of MD membranes using seven PVDF membranes synthesized under different conditions. The following conclusions were drawn based on the findings:Depending on the fabrication conditions, membranes with flux, rejection, contact angle (CA), liquid entry pressure (LEP), and pore sizes were obtained. Without LiCl, a membrane with small pore size, high LEP and low flux was prepared. When LiCl was used, an increase in PVDF concentration led to the formation of denser membranes. The flux and rejection were further adjusted by controlling the EtOH concentration in the non-solvent.Using the equations derived in this work, *B_w_*, *L_w_*, and *R_f_* were calculated. It was found that *B_w_* and *L_w_* ranged from 7.76 × 10^−8^ s/m to 20.19 × 10^−8^ s/m and from 0.0020 to 0.0151, respectively. An increase in *B_w_* resulted in an increased *L_w_*, indicating a trade-off between the two properties. This is attributed to the fact that the membranes with higher *B_w_* values have larger pore sizes.*R_f_* was calculated in the range from 0.598 μm to 1.690 μm. Since the *R_f_* was relatively small, the prepared membranes can have high LEP (more than 1.13 bar) even at low CA (less than 90.8°). *R_f_* was found to be correlated with the surface roughness measured by AFM.An efficient MD membrane should have a high flux, rejection, and LEP with low fouling propensity. The results in this study suggest that the pore size should be high to ensure high *B_w_* but *R_f_* should be small to lower *L_w_*. However, care should be taken in this approach. Since there is a trade-off between pore size and *R_f_*, it may not be possible to simultaneously increase both properties. In addition, an increase in the pore size above a critical value is not allowed due to high risk of the wetting.If *R_f_* is sufficiently small, it is plausible to fabricate membranes using moderately hydrophobic materials, which is beneficial to retard fouling due to hydrophobic foulants. Nevertheless, further work should be done to examine this hypothesis.

## Figures and Tables

**Figure 1 membranes-11-00437-f001:**
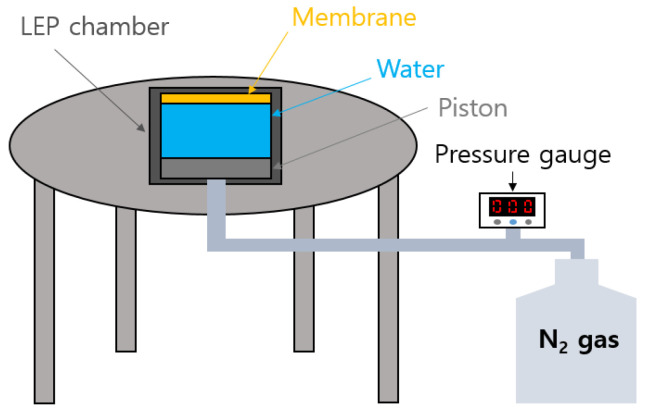
Schematic drawing of LEP apparatus.

**Figure 2 membranes-11-00437-f002:**
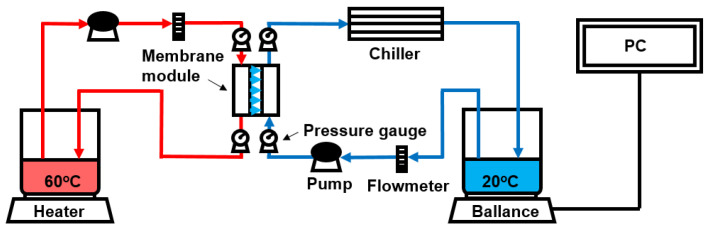
Schematic diagram of an equipment for DCMD experiments.

**Figure 3 membranes-11-00437-f003:**
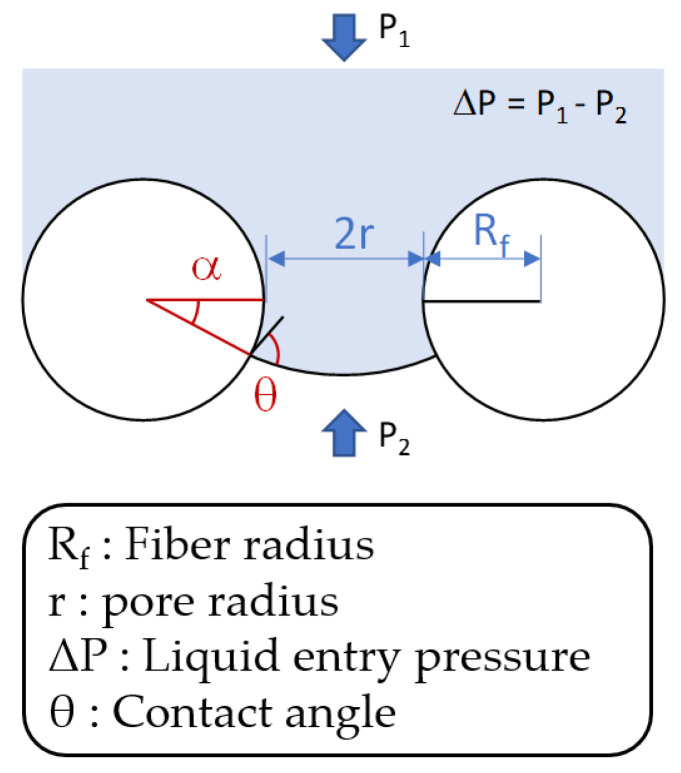
Interface in toroidal pore of hydrophobic membrane based on Purcell model.

**Figure 4 membranes-11-00437-f004:**
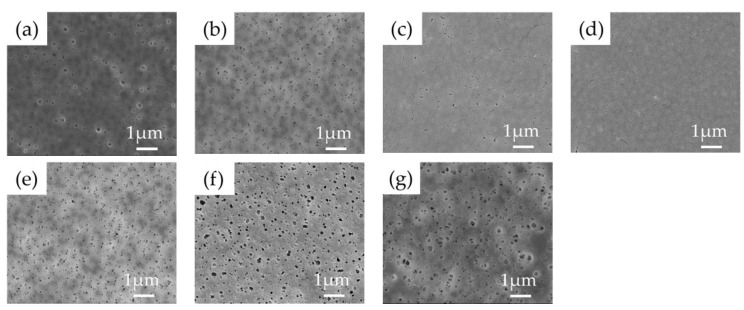
FE-SEM images of top surface of flat-sheet membranes with different PVDF, LiCl concentration and non-solvent compositions: (**a**) (S1) PVDF 14 wt.% with water, (**b**) (S2) PVDF 14 wt.% + LiCl with water, (**c**) (S3) PVDF 16 wt.% + LiCl with water, (**d**) (S4) PVDF 18 wt.% + LiCl with water, (**e**) (S5) PVDF 16 wt.% + LiCl with water (90%) + EtOH (10%), (**f**) (S6) PVDF 16 wt.% + LiCl with water (80%) + EtOH (20%) and (**g**) (S7) PVDF 16 wt.% + LiCl with water (70%) + EtOH (30%).

**Figure 5 membranes-11-00437-f005:**
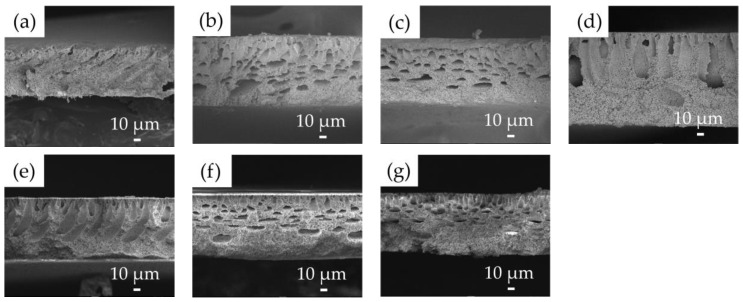
FE-SEM images of cross-section of flat-sheet membranes with different PVDF, LiCl concentration and non-solvent compositions: (**a**) (S1) PVDF 14 wt.% with water, (**b**) (S2) PVDF 14 wt.% + LiCl with water, (**c**) (S3) PVDF 16 wt.% + LiCl with water, (**d**) (S4) PVDF 18 wt.% + LiCl with water, (**e**) (S5) PVDF 16 wt.% + LiCl with water (90%) + EtOH (10%), (**f**) (S6) PVDF 16 wt.% + LiCl with water (80%) + EtOH (20%) and (**g**) (S7) PVDF 16 wt.% + LiCl with water (70%) + EtOH (30%).

**Figure 6 membranes-11-00437-f006:**
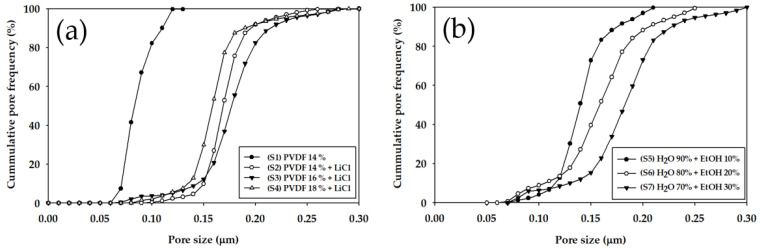
Cumulative pore frequency versus pore size of the membranes with different PVDF, LiCl concentration and non-solvent compositions: (**a**) (S1) PVDF 14 wt.% with water, (S2) PVDF 14 wt.% + LiCl with water, (S3) PVDF 16 wt.% + LiCl with water, and (S4) PVDF 18 wt.% + LiCl with water; (**b**) (S5) PVDF 16 wt.% + LiCl with water (90%) + EtOH (10%), (S6) PVDF 16 wt.% + LiCl with water (80%) + EtOH (20%), and (S7) PVDF 16 wt.% + LiCl with water (70%) + EtOH (30%).

**Figure 7 membranes-11-00437-f007:**
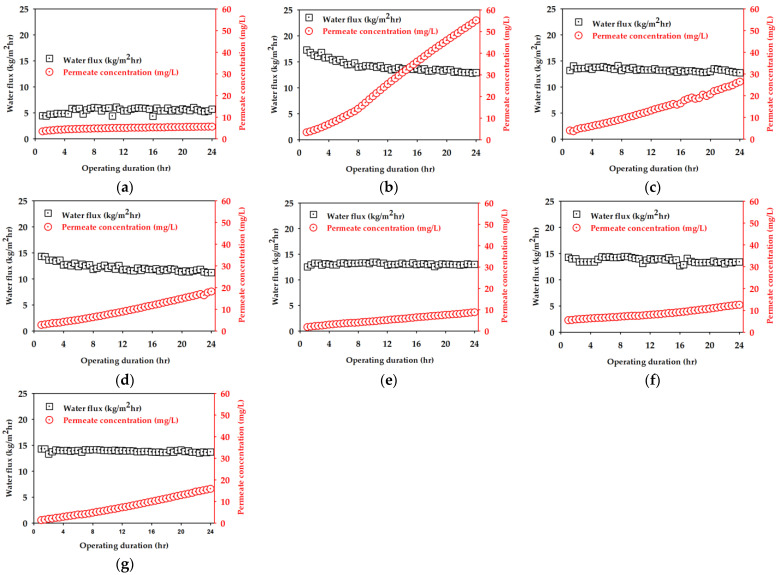
Water flux and salt permeate concentration for 24 h operation by DCMD on 35 g/L NaCl solution: (**a**) (S1) PVDF 14 wt.% with water, (**b**) (S2) PVDF 14 wt.% + LiCl with water, (**c**) (S3) PVDF 16 wt.% + LiCl with water, (**d**) (S4) PVDF 18 wt.% + LiCl with water, (**e**) (S5) PVDF 16 wt.% + LiCl with water (90%) + EtOH (10%), (**f**) (S6) PVDF 16 wt.% + LiCl with water (80%) + EtOH (20%), and (**g**) (S7) PVDF 16 wt.% + LiCl with water (70%) + EtOH (30%). For all DCMD experiments, the feed and distillate temperatures were 60 and 20 °C, respectively.

**Figure 8 membranes-11-00437-f008:**
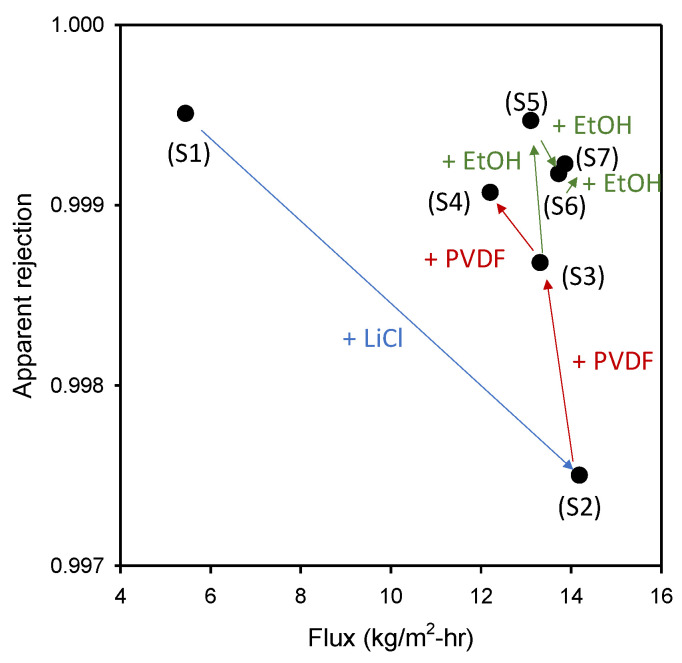
Changes in flux and apparent rejection with membrane fabrication conditions: An overview.

**Figure 9 membranes-11-00437-f009:**
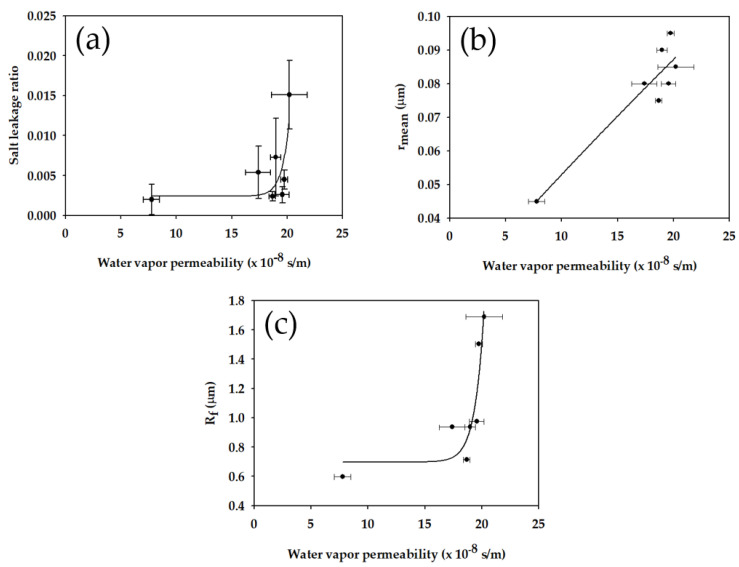
Effect of water vapor permeability (*B_w_*) on (**a**) salt leakage ratio (*L_w_*); (**b**) mean pore radius; (**c**) fiber radius (*R_f_*) of fabricated membranes.

**Figure 10 membranes-11-00437-f010:**
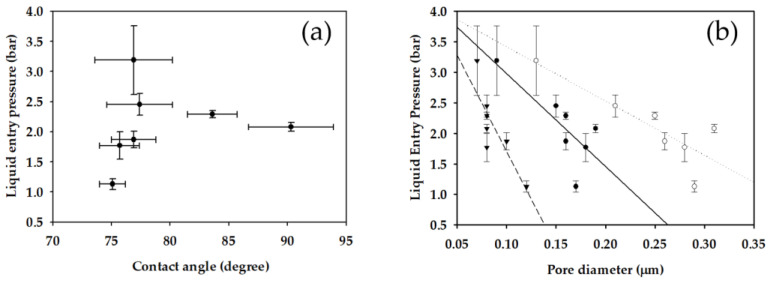
Exploration of factors affecting LEP of fabricated membranes: (**a**) Contact angle; (**b**) pore diameters (▼: minimum pore diameter; ●: mean pore diameter; ○: maximum pore diameter).

**Figure 11 membranes-11-00437-f011:**
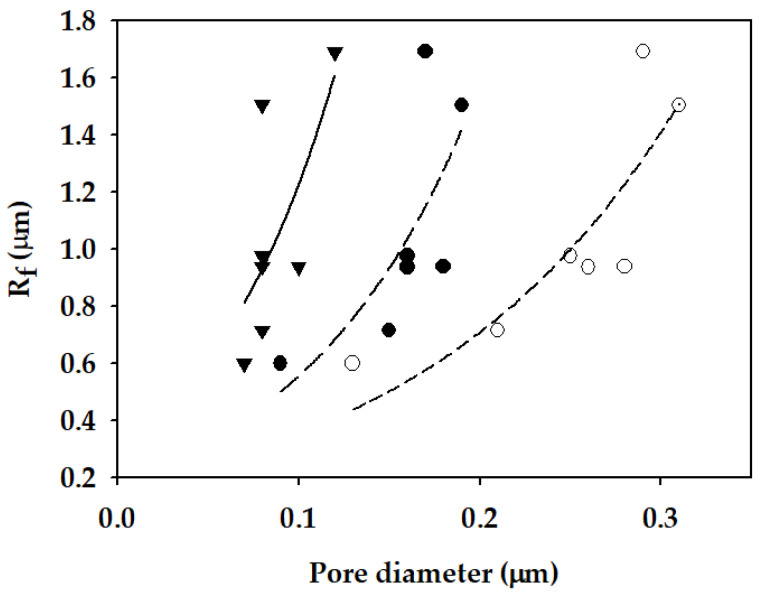
Dependence of *R_f_* on pore sizes of fabricated membranes (▼: minimum pore diameter; ●: mean pore diameter; ○: maximum pore diameter).

**Figure 12 membranes-11-00437-f012:**
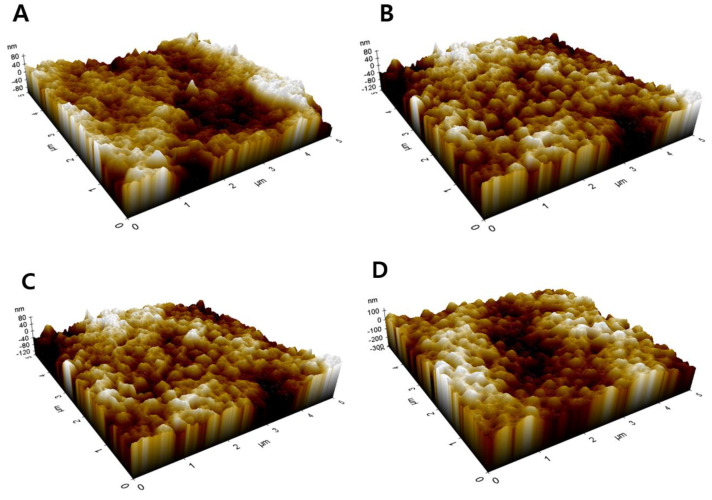
AFM images of fabricated membranes: (**A**) (S3) Water; (**B**) (S5) Water (90%) + EtOH (10%); (**C**) (S6) Water (80%) + EtOH (20%); and (**D**) (S7) Water (70%) + EtOH (30%).

**Figure 13 membranes-11-00437-f013:**
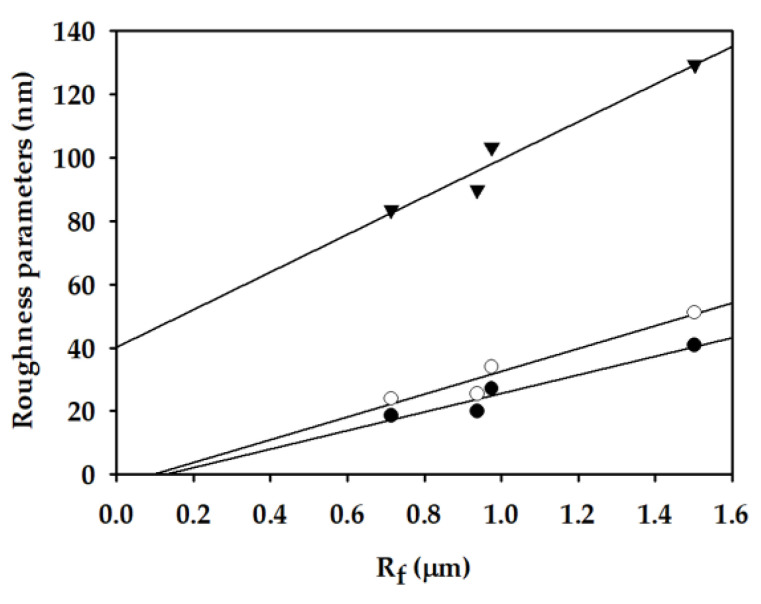
Roughness parameters versus Fiber radius of fabricated membranes (●: arithmetical mean deviation (*R_a_*); ○: root mean square deviation (*R_q_*); ▼: vertical distance between highest peak and lowest valley (*R_max_*).

**Table 1 membranes-11-00437-t001:** Experimental conditions for membrane fabrication.

Membrane Sample	PVDF Concentration [*w*/*w*]	LiCl Concentration [*w*/*w*]	Solvent	Non-Solvent [*v*/*v*]
S1	14.0%	0.0%	DMF	Water (100%)
S2	14.0%	3.0%	DMF	Water (100%)
S3	16.0%	3.0%	DMF	Water (100%)
S4	18.0%	3.0%	DMF	Water (100%)
S5	16.0%	3.0%	DMF	Water (90%) + EtOH (10%)
S6	16.0%	3.0%	DMF	Water (80%) + EtOH (20%)
S7	16.0%	3.0%	DMF	Water (70%) + EtOH (30%)

**Table 2 membranes-11-00437-t002:** Primary (measurable) properties of fabricated membranes.

Membrane Sample	Contact Angle (°)	LEP (bar)	Membrane Thickness (μm)	Porosity (%)	*d_mean_* (μm)	*d_max_* (μm)	*d_min_* (μm)
S1	76.9 ± 3.3	3.19 ± 0.57	68.0 ± 3.4	85.4 ± 2.5	0.09	0.13	0.07
S2	75.1 ± 1.1	1.13 ± 0.09	80.3 ± 3.3	88.3 ± 2.3	0.17	0.29	0.12
S3	75.7 ± 1.7	1.77 ± 0.23	85.2 ± 6.4	85.2 ± 2.0	0.18	0.28	0.08
S4	76.9 ± 1.9	1.87 ± 0.14	98.8 ± 3.0	84.9 ± 1.3	0.16	0.26	0.10
S5	77.4 ± 2.8	2.45 ± 0.18	78.3 ± 2.4	86.1 ± 4.2	0.15	0.21	0.08
S6	83.6 ± 2.1	2.29 ± 0.06	79.7 ± 7.5	81.9 ± 1.5	0.16	0.25	0.08
S7	90.3 ± 3.6	2.08 ± 0.07	76.8 ± 0.8	83.6 ± 1.1	0.19	0.31	0.08

**Table 3 membranes-11-00437-t003:** Average water flux and apparent rejection for 24 h operation by DCMD on 35 g/L NaCl solution.

Membrane Sample	Flux (kg/m^2^-h)	Apparent Rejection (%)
S1	5.45 ± 0.50	99.95 ± 0.005
S2	14.19 ± 1.14	99.75 ± 0.146
S3	13.32 ± 0.33	99.87 ± 0.061
S4	12.21 ± 0.78	99.91 ± 0.0043
S5	13.11 ± 0.19	99.95 ± 0.019
S6	13.73 ± 0.46	99.92 ± 0.0190
S7	13.87 ± 0.21	99.93 ± 0.041

**Table 4 membranes-11-00437-t004:** Secondary (evaluated) properties of fabricated membranes.

Membrane Sample	Water Vapor Permeability, *B_w_* (×10^−8^ s/m)	Salt Leakage Ratio, *L_w_* (-)	Fiber Radius, *R_f_* (μm)
S1	7.76 ± 0.71	0.0020 ± 0.0019	0.598
S2	20.19 ± 1.62	0.0151 ± 0.0043	1.690
S3	18.96 ± 0.47	0.0073 ± 0.0049	0.938
S4	17.38 ± 1.11	0.0054 ± 0.0033	0.936
S5	18.66 ± 0.27	0.0024 ± 0.0006	0.714
S6	19.54 ± 0.65	0.0026 ± 0.0010	0.975
S7	19.74 ± 0.30	0.0045 ± 0.0012	1.503

**Table 5 membranes-11-00437-t005:** Roughness parameters of the prepared membranes.

Membrane Sample	*R_a_* (nm) ^a^	*R_q_* (nm) ^b^	*R_max_* (nm) ^c^
S3	20.01	25.47	89.87
S5	18.51	23.97	83.49
S6	27.03	34.09	103.22
S7	40.80	51.13	129.44

^a^*R_a_* (nm): Arithmetical mean deviation. ^b^
*R_q_* (nm): Root mean square deviation. ^c^
*R_max_* (nm): Vertical distance between highest peak and lowest valley.

**Table 6 membranes-11-00437-t006:** Comparison of the membrane properties obtained in this study with the literature for DCMD process.

Membrane Sample	Contact Angle (°)	LEP (bar)	*d_mean_* (μm)	Flux (kg/m^2^h)	Apparent Rejection (%)	Water Vapor Permeability, *B_w_*, (×10^−8^ s/m)	Salt Leakage Ratio, *L_w_* (-)	Fiber Radius, *R_f_* (μm)
S6	83.6 ± 2.1	2.29 ± 0.06	0.16	13.73 ± 0.46	99.92 ± 0.0190	19.54 ± 0.65	0.0026 ± 0.0010	0.975
S7	90.3 ± 3.6	2.08 ± 0.07	0.19	13.87 ± 0.21	99.93 ± 0.0410	19.74 ± 0.30	0.0045 ± 0.0012	1.503
M1 [40]	75.7 ± 1.4	2.93 ± 0.06	0.11	20.20 ± 0.10	99.93 ± 0.0182	28.73 ± 0.27	0.0015 ± 0.0003	0.618
M2 [40]	73.2 ± 2.7	4.16 ± 0.25	0.07	8.6 ± 0.11	99.93 ± 0.0145	12.26 ± 0.12	0.0026 ± 0.0007	0.350
Commercial PVDF [40]	126.8 ± 1.1	1.81 ± 0.16	0.22	15.10 ± 0.61	99.93 ± 0.0087	21.69 ± 0.65	0.0023 ± 0.0009	0.429

## Data Availability

Not applicable.

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
