# Peer review of "Analysis of Polyvinylidene Fluoride Membranes Fabricated for Membrane Distillation"

_membranes, 2021, doi:10.3390/membranes11060437_

Round 1
Reviewer 1 Report
The manuscript deals with PVDF asymmetric membranes for MD. Using commercial PVDF, the membranes were fabricated by phase inversion in different conditions. Many PVDF membranes reported for MD have nanofibrous morphology for better hydrophobicity and higher vapor permeability, but the asymmetric membranes in this study will have advantages for up-scaling for industrial use. Experimental procedure was described in detail. I recommend minor revision for accepting for publication in Membranes.
- PVDF is one of the most common membrane materials, so the properties of the membranes should be compared with other PVDF membranes reported. Otherwise, it would be good to test commercial PVDF membrane and directly compare with other membranes (S1-S7).
- Figure 1 and Figure 4a are not necessary diagrams that can be just described in sentences.
- Scale bars in Figure 5 are not visible that need to be bigger. Figure 5 has too many images that it needs to be separated into two (might be surface and cross-section?). SEM images can show pore diameter roughly, but the images in Figure 5 are too small. I don’t feel Figure 5o (photo) is not necessary.
- Captions in Figures are too small. The numbers are not readable. This font should be bigger.
- The authors mentioned that fouling is an important issue for MD process to be considered. What types of foulant the feed solution has in practical MD applications? How are you sure that the PVDF membranes in this study can have advantages against fouling?
- Contact angles of the membranes are between 75 to 90 degree, meaning that the membranes are rather hydrophilic. Do the membranes have wetting issue? Can you add the degree of wetting?
- MD performances in Figure 7 might the most important, but the explanation is too brief. Membranes S3-S7 seem to have similar performance although they are prepared in different conditions having different properties listed in table 2. S2 has the highest ratio in increase of permeate concentration. What are reasons?
- What does the fibre radius mean? Fibre radius might be applied for nanofibre membranes, but asymmetric membranes don’t have fibres, only continuous asymmetric morphology unless having microfibres inside.
Author Response
We are grateful for your valuable comment.
We have modified this paper as your comments.
Please see the attachment.

Reviewer 2 Report
The manuscript is well written and well organized and understandable. I recommend it for publication.
Please, read the comments below:
- The introduction lacks detailed information on the fouling and scaling phenomena. Authors should present in more detail information about:
- the impact of fouling and scaling on the MD process efficiency,
- methods of minimizing their intensity. - Line 42: Percent sign and the number should not separated by space.
- Why did the Authors write '35,000 mg/L 'with a to three decimal places?
Author Response
We are grateful for your valuable comment.
We have modified this paper as your comments.
Please see the attachment.

This manuscript is a resubmission of an earlier submission. The following is a list of the peer review reports and author responses from that submission.
Round 1
Reviewer 1 Report
The manuscript by Lee and co-workers report the synthesis of PVDF membranes by phase inversion method. Furthermore, the authors examine the properties of membrane and correlate with the synthetic strategy. This work is of interest to the readers and could be suitable for Membrane after consideration of the following points:
- The authors should illustrate in detail the role of ethanol in membrane fabrication? What will happen if ethanol is substituted by other protic or aprotic solvent or if the ratio of water : ethanol ratio is altered to 1:1.
- The numbering of figures should be revised. Furthermore, from figure 6-9, insert the alphabets in the figures similar to figure 5.
Author Response
We are grateful for your valuable comments.
I have modified this paper as your comments.
Please see the attachment.
Best regards.

Reviewer 2 Report
In the manuscript entitled “Systematic Analysis of Polyvinylidene Fluoride Membranes 2 Fabricated for Membrane Distillation” authors present their methodology for correlating membrane intrinsic properties with performance in direct contact membrane distillation.
After reviewing the document I have the following comments and recommendations:
The authors base their conclusion in a very limited amount of membranes. In my opinion these cannot qualify as a basis for a systematic analysis
The authors make connections between properties that are not physically linked. For example LEP and permeability are connected with a line in Figure 10; do the authors suggest that an equation exist to connect these 2 factors?
The experimental process is not well presented, the transmembrane pressure is not monitored while having different fluxes between cold and hot streams.
Figure 7 is confusing: The graph presents the permeate concentration while I presume that the conductivity presented is of the permeate tank. This is a misleading.
The manuscript does not present novel data , makes no comparison with existing data and my opinion is that in its present form cannot be accepted for publication.
Author Response

(The authors gave the same response as above.)

Reviewer 3 Report
The present paper deals with the application of a systematic, new and original, approach, in MD elaboration filed of research.
We can suggest few little change to improve the submitted paper :
- lInes 99 and 470 : add micrometer to m
- I can suggest to add in the article a photography of 1 or all flat sheet membranes elaborated
- I don't understand Rf parameter; do you well elaborated flat sheet membranes ? If yes, I don't understand fiber length Rf what does it means ? Is it similar to pore size diameter ?
- As reported in literature, a typical MD must present the following
properties: a contact angle higher than 90°, see 2017 Desalination 423 (2017) 30–40 (and the references therin). Ass this reference to your reference list pleased.
Concerning the contact angle values reported you are under the usual MD values observed elsewhere and suggested to attain; could your explain this fact please and how to improve it.
Furthermore, as you know real seawater could create an hydrophilisation of the membrane surface how can you protect the membrane agains this real experiments consequances ?
Considering the quality and originality of the present paper I suggest MINOR REVISIONS.
Author Response

(The authors gave the same response as above.)

Round 2
Reviewer 2 Report
The authors continue to correlate LEP with membrane properties such as permeability. This in my opinion is a major flaw of the manuscript and as such I do not recommend this study to be accepted.